# Tuning Physicochemical Properties of a Macroporous Polysaccharide-Based Scaffold for 3D Neuronal Culture

**DOI:** 10.3390/ijms222312726

**Published:** 2021-11-25

**Authors:** Gaspard Gerschenfeld, Rachida Aid, Teresa Simon-Yarza, Soraya Lanouar, Patrick Charnay, Didier Letourneur, Piotr Topilko

**Affiliations:** 1Ecole Normale Supérieure, PSL Research University, CNRS, Inserm, Institut de Biologie de l’Ecole Normale Supérieure (IBENS), F-75005 Paris, France; gaspard.gerschenfeld@aphp.fr (G.G.); charnay@bio.ens.psl.eu (P.C.); 2Collège Doctoral, Sorbonne Université, F-75005 Paris, France; 3INSERM U1148, LVTS, Université de Paris, X Bichat Hospital, 46 Rue H Huchard, F-75018 Paris, France; rachida.aid@inserm.fr (R.A.); teresa.simon-yarza@inserm.fr (T.S.-Y.); soraya.lanouar@gmail.com (S.L.); didier.letourneur@inserm.fr (D.L.); 4INSERM UMS-34, FRIM, Université de Paris, X Bichat School of Medicine, F-75018 Paris, France; 5INSERM U1148, LVTS, Université Sorbonne Paris Nord, 99 Av JB Clément, F-93430 Villetaneuse, France; 6Institut Mondor de Recherche Biomédicale (IMRB), Université Paris Est Créteil (UPEC), INSERM U955, F-94010 Créteil, France

**Keywords:** porous scaffold, polysaccharide, pullulan–dextran, embryonic neurons

## Abstract

Central nervous system (CNS) lesions are a leading cause of death and disability worldwide. Three-dimensional neural cultures in biomaterials offer more physiologically relevant models for disease studies, toxicity screenings or in vivo transplantations. Herein, we describe the development and use of pullulan/dextran polysaccharide-based scaffolds for 3D neuronal culture. We first assessed scaffolding properties upon variation of the concentration (1%, 1.5%, 3% *w*/*w*) of the cross-linking agent, sodium trimetaphosphate (STMP). The lower STMP concentration (1%) allowed us to generate scaffolds with higher porosity (59.9 ± 4.6%), faster degradation rate (5.11 ± 0.14 mg/min) and lower elastic modulus (384 ± 26 Pa) compared with 3% STMP scaffolds (47 ± 2.1%, 1.39 ± 0.03 mg/min, 916 ± 44 Pa, respectively). Using primary cultures of embryonic neurons from *PGK^Cre^, Rosa26^tdTomato^* embryos, we observed that in 3D culture, embryonic neurons remained in aggregates within the scaffolds and did not attach, spread or differentiate. To enhance neuronal adhesion and neurite outgrowth, we then functionalized the 1% STMP scaffolds with laminin. We found that treatment of the scaffold with a 100 μg/mL solution of laminin, combined with a subsequent freeze-drying step, created a laminin mesh network that significantly enhanced embryonic neuron adhesion, neurite outgrowth and survival. Such scaffold therefore constitutes a promising neuron-compatible and biodegradable biomaterial.

## 1. Introduction

Central nervous system (CNS) lesions are a leading cause of disability worldwide due to their limited regeneration potential [1,2] Numerous three-dimensional (3D) neural cell culture systems within biomaterials, mimicking the CNS extracellular environment, are being developed for drug screening studies [3,4] and as scaffolds in tissue engineering approaches [5,6]. A suitable biomaterial for neural culture should have a porous structure with mechanical properties similar to the CNS; should support neural cell survival, adhesion and neurite outgrowth; and should be biodegradable as well as biocompatible [7]. Hydrogels are 3D hydrated porous polymer networks whose physicochemical properties can be tailored with interconnected pores that can allow for extensive neurite extension before they degrade [8,9,10]. Natural polymers such as collagen [11], chitosan [12], gelatin, hyaluronic acid [13], alginate [14] and heparan sulfate [15] have retained some interest as scaffolding biomaterial for neural tissue engineering because of their superior biological compatibility compared with synthetic scaffolds [7,10,16]. However, once transplanted in the CNS, natural polymers can induce specific foreign body responses that prevent their integration with the host tissue [17].

We have recently developed a hydrogel based on two nonimmunogenic naturally derived polymers: pullulan and dextran. Pullulan is a linear polysaccharide produced from starch fermentation by the fungus *Aureobasidium pullulan*, which consists of glucose units linked through α1,6- and α1,4-glucosidic bonds, whose branched polymeric structure offers flexibility. Dextran is a polysaccharide synthesized from sucrose by bacteria, which is composed of glucose units joined mostly by α1,6-glycosidic bonds, with α1,2-, α1,3- or α1,4- side chains, which provides structural stability thanks to its high molecular weight [18,19,20]. Both are hydrophilic and neutral polymers biochemically similar to the extracellular matrix (ECM) that can be degraded by enzymes present in most mammalian tissues [21]. Both polymers can also be functionalized to adapt their physical and biological properties depending on specific needs [22]. Interestingly, crosslinking pullulan with dextran reduces the foreign body response following transplantation compared with a dextran-only material [23]. Moreover, they can be produced under GMP-conditions at a large-scale and have been successfully implanted in mice, rats, pigs and goats without inducing any major inflammatory response [21,23,24]. To conceive porous polysaccharide-based hydrogels, we developed a freeze-drying cross-linking process based on sodium trimetaphosphate (STMP), a simple and non-toxic method for producing scaffolds easily at low cost and without organic solvent. Thereafter, we functionally validated these polysaccharide-based scaffolds in several medical applications such as vascular grafts [21], endothelial cell [25], vascular smooth muscle cell [19], hematopoietic stem cell [26] or hepatic organoids culture [27], mesenchymal stem cell delivery into myocardial infarction [28] and bone lesion repair [29], but its use to manipulate neurons in 3D has not been explored to date.

The aim of this study was to tailor the physicochemical properties of a macroporous polysaccharide-based scaffold to improve its ability to promote the attachment and neurite outgrowth of mouse embryonic neurons.

## 2. Results

### 2.1. Hydrogel Structure and Porosity Are Correlated with STMP Concentration

Three-dimensional porous polysaccharide-based scaffolds were prepared by a chemical cross-linking process with STMP and sodium carbonate as a porogen agent and then freeze-dried. Three formulations were prepared with 1%, 1.5% and 3% (*w*/*w*) STMP concentrations and were referenced as 1%, 1.5% and 3% matrices, respectively. Analysis by scanning electron microscopy (SEM) of scaffold surfaces and sections confirmed the presence of pores with a smooth and homogeneous wall surface in the three formulations (Figure 1a–f). Gross scaffold morphology varied depending on STMP concentration, with the surface of 3% matrices that appeared denser and smoother. Porosity analysis based on the pore area and the total scaffold area ratio was obtained from confocal acquisitions using FITC-Dextran in the formulations. While there was no significant difference in porosity between 1% and 1.5% matrices (59.9 ± 4.6% and 58.8 ± 4%, respectively), 3% matrices (47 ± 2.1%) displayed the lowest porosity. Since cross-linking is obtained through phosphate bridge formation between polysaccharide chains, the cross-linking ratio can be assessed by the phosphate content. Scaffold phosphate content quantified by a colorimetric assay was positively correlated with the introduced STMP concentrations (Table 1). Indeed, 3% matrices had significantly more phosphate content (236 ± 7 μmol/g of dried scaffold) than 1.5% matrices (108 ± 5 μmol/g) and 1% matrices (84 ± 1 μmol/g).

### 2.2. Swelling Behavior and Rheological Analysis

Next, we measured the swelling kinetics of each formulation before and during the 24 h following their hydration in PBS and in water (Figure 1g). The swelling and absorbing abilities of hydrogels are crucial for several functions: (i) the cell seeding process that is based on the absorption of a cell suspension; (ii) the exchange of substances (nutrients, oxygen and wastes) during cell culture; and (iii) potentially in the optic of drug delivery applications. We investigated it in both water and PBS to assess whether it is dependent on ionic strength, as their ionic strength difference is important (the conductivity of water is around 15 µSi/cm vs. 15 mSi/cm for the PBS), and because PBS is similar to cell culture medium in terms of conductivity, allowing better prediction of swelling under in vitro conditions. Matrices prepared with 1%, 1.5% and 3% (*w*/*w*) STMP swelled rapidly in both water and PBS and reached a steady state in less than two hours. A significant difference was noted between the swelling in PBS and in water for the three formulations, the swelling being higher in water. In PBS, the swelling ratio was significantly higher for 1% and 1.5% matrices than for 3%. A similar significant difference was observed in water for 1%, 1.5% and 3%. These results allow us to conclude that the hydrogel swelling is inversely correlated to the ionic strength of the solvent and the concentration of the crosslinking agent.

Mechanical properties of 1%, 1.5% and 3% matrices were also investigated by rheological measurements (Figure 2). Storage G′ and loss G″ moduli average of all conditions are given at a 1 Hz frequency. For each formulation, G′ was more than one order of magnitude higher than G″. Both moduli G′ and G″ increased with the increase in STMP concentration. The elastic modulus measured in 3% matrices was 1.4 times higher than in 1.5% matrices (916 ± 44 Pa and 639 ± 10, respectively) and 2.4 times higher than 1% matrices (384 ± 26 Pa). Hence, hydrogel stiffness was positively correlated with the crosslinking ratio.

### 2.3. Hydrogel Structure and Porosity Are Correlated with STMP Concentration

To measure the degradation kinetics, polysaccharide-based scaffolds were degraded in vitro using a mixture of pullulanase and dextranase, which digest pullulan and dextran, respectively. Measurements were made with the three formulations (matrices 1%, 1.5% and 3%) every ten minutes, and degradation was determined by wet weight loss (Figure 1h). STMP concentration was positively correlated with the degradation time. Indeed, the 3% matrix was degraded in 70 min, while complete degradation of the lower cross-linked scaffolds, 1.5% and 1% matrix, was reached after 25 and 20 min, respectively. Degradation rates expressed in mg/min (Table 1) calculated from the degradation curves were thus higher in 1% and 1.5% matrix compared with 3% matrix (5.11 ± 0.14 and 3.83 ± 0.12 vs. 1.39 ± 0.03, respectively).

### 2.4. Embryonic Neurons Aggregate within the Macroporous Scaffolds

To assess the polysaccharide-based scaffolds biocompatibility with neuronal culture, we seeded embryonic neurons on 1%, 1.5% and 3% matrices and cultured them for up to four days. After two days of culture, neurons formed aggregates within the pores of each type of scaffold, despite of scaffold morphological differences (Figure 3). Higher magnification imaging showed that while neurons regrouped in aggregates, they did not attach directly to the scaffold whatever the STMP content, and there were no neurite extensions under these culture conditions.

### 2.5. Functionalized Scaffolds Preparation and Characterization

In order to improve neuronal adhesion, we selected 1% matrices, whose stiffness (384 ± 26 Pa) was within the optimal range for neuronal culture between 100 and 500 Pa [10,29,30]. We prepared two conditions: (i) simple hydration with a laminin solution (100 μg/mL) during cell seeding (Matrix 1% L); or (ii) hydration with laminin and freeze-dried before cell seeding (Matrix 1% L-FD) to better keep the laminin within the polymeric network. SEM analysis confirmed that Matrix 1% L-FD scaffolds displayed a similar morphology to the Matrix 1% formulation but also revealed a mesh network of laminin around and within the scaffold (Figure 4). Compared with Matrix 1%, Matrix 1% L-FD scaffolds did not differ in terms of phosphate content (84 ± 1 vs. 86 ± 2 µmol/g) or swelling ratio (47.1 ± 6 vs. 46.6 ± 6 in PBS and 123 ± 16 vs. 118 ± 12 in water).

### 2.6. Laminin Functionalization Promotes Cellular Attachment within the Scaffold

To assess the impact of laminin functionalization of 1% matrices, embryonic neurons were seeded on: (i) Matrix 1% without laminin, (ii) Matrix 1% with laminin (100 μg/mL) simply added in the cell suspension before cell seeding (Matrix 1% L), and (iii) Matrix 1% L-FD with impregnation and freeze drying (Figure 5). After two days of culture, embryonic neurons seeded on Matrix 1% L-FD scaffolds formed numerous large cell aggregates, extended neurites on and within the scaffold and maintained the expression of the immature neuron marker βIII-tubulin (Tuj1, Figure 5d,f). Such phenotype was observed in the Matrix 1% L (simple addition) but at a much lower extent (Figure 5 middle panel).

To further characterize the effect of laminin functionalization on embryonic neuron behavior, we measured the mean cell number within the three types of scaffolds after one, four and seven days of culture. While the mean cell number per scaffold was similar at day 1—around 10^4^ cells per scaffold in Matrix 1% (10,693), Matrix 1% L (10,782) and Matrix L-FD (8645)—it decreased after one week in the Matrix 1% (4114) and Matrix 1% L (4400), but remained stable in Matrix 1% L-FD (8600).

## 3. Discussion

In this study, we described the development and optimization of polysaccharide-based biomaterial for 3D neuronal culture. First, we optimized the STMP concentration, which allowed the generation of soft scaffolds with a high porosity. In water and PBS, the resulting hydrogels were stable and swelled. From all three formulations, we were able to generate 3D porous scaffolds that allowed cell infiltration within a few minutes. Interestingly, Matrix 1% scaffolds had an elastic modulus of 384 ± 26 Pa, which is within the reported optimal stiffness range for neuronal culture, between 100 and 500 Pa [10,30,31,32]. This may be of critical importance, as stiffness impacts neuronal growth and neurite extensions [33]. The obtained matrices at 1% STMP also exhibited faster degradation rates that could be more suitable for in vivo applications, facilitating the engraftment of transplanted cells.

In our work, we chose to use primary cultures of embryonic neurons from *PGK^Cre^, Rosa26^tdTomato^* embryos for three reasons: (i) this technique is commonly used to generate long-term cultures of mature neurons [34]; (ii) the use of the tdTomato, a membranous fluorescent reporter, enables the morphological live imaging of all seeded cells during cell culture with an epi-fluorescent macroscope [35]; (iii) in the long term, tdTomato embryonic neurons could be of interest in the hypothesis of transplantation experiments in a tissue engineering approach. The fact that embryonic neurons neither significantly attached nor extended neurites in non-functionalized scaffolds was not surprising, as the need for additional adhesion molecules has been largely reported [5]. We thus decided to add laminin in our scaffolds for two reasons: (i) it is one of the major ECM components in the brain; (ii) it is known to promote neuronal adhesion in 2D and 3D cultures [5,36,37,38]; and (iii) it has been shown to enhance the proliferation and differentiation of neuronal stem cells in 3D cultures [39]. We also explored several approaches to attach laminin to the scaffold. When we added the laminin in the pullulan/dextran mixture before the cross-linking, laminin was denatured. When laminin was simply added during cell seeding (Matrix 1% L), it yielded poor cell adhesion and neurite outgrowth. In contrast, when laminin was added and then freeze-dried, it created a mesh network of laminin within and on the surface of the scaffold. This technique has several advantages, as it does not require additional cross-linking steps, it is a non-toxic process and it allows scaffolds to be stored at room temperature over several months. The enhanced neuronal adhesion and neurite outgrowth observed with the Matrix 1% L-FD confirmed that this formulation was optimized for 3D culture (Figure 5). Altogether, to our knowledge, our study provides the first demonstration that hydrogels made of pullulan and dextran are suitable for neuronal culture. We have shown that the way in which laminin is incorporated in the formulation has a major impact on neuronal cell adhesion and neurite outgrowth. Compared with other studies where chemical modifications of polymers were used [14,40,41], here laminin was deposited on the pores of the scaffold during the freeze-drying step, reducing the number of chemical reagents and synthesis steps, and leading to an off-the-shelf scaffold. Interestingly, this formulation could also be used in clinical applications since both the manufacturing of the scaffolds, which is in compliance with the good manufacturing practice (GMP) guidelines, and the use of clinical-grade laminin have been validated.

Pullulan/dextran-L-FD scaffolds present numerous advantages both in vitro and in vivo: (i) they are made of polysaccharides and laminin and are easy to produce, potentially in compliance with GMP guidelines; (ii) they can be stored dried at room temperature for long periods without affecting their properties; (iii) they can be produced in different shapes and sizes—for instance, we used 350 µm thick and 5 mm wide disks that could be cultured in separate wells and handled easily using a spatula; (iv) they are transparent, which enables direct microscopic observation during cell culture and facilitates immunostainings and confocal imaging. Hence, these scaffolds could be used in several types of applications, ranging from long-term in vitro 3D cell culture for disease modelling or drug toxicity screening to in vivo application, either as a cell-seeded scaffold or an acellular scaffold for tissue engineering.

While an embryonic neuron-seeded scaffold is an attractive model, this approach has some limitations. First, it has been shown that for proper neuronal integration, grafted neurons need to be very precisely differentiated into the neuron type that they are replacing [42,43]. Second, to be more clinically relevant, human neural cell lines should be tested with the addition of glial cells, as they play several key roles in brain function. Third, depending on the scaffold thickness, the initial lack of vascularization could lead to the death of transplanted cells. One way to overcome this limitation could be by adding another polysaccharide in the formulation, for instance fucoidan, which has the ability to sequester and release vascular endothelial growth factor (VEGF), promoting neovascularization in vivo [44] or to create a preformed vascular network within the hydrogel [45].

## 4. Materials and Methods

### 4.1. Porous Scaffold Synthesis

Polysaccharide-based scaffolds were prepared using a mixture of pullulan/dextran in water [19,46,47] (pullulan, MW 200 kDa, Hayashibara; dextran, MW 500 kDa, Pharmacosmos). A portion of 40% sodium carbonate (*w*/*w*) was added to the mixture as a porogen agent. Polysaccharides chemical cross-linking was carried out using three concentrations of 1%, 1.5% and 3% sodium trimetaphosphate (STMP) in 1M sodium hydroxide, referenced as Matrix 1%, 1.5% and 3%, respectively. Scaffolds were incubated at 50 °C for 15 min, then cut into the desired shape (5 mm diameter, 350 µm thickness) before being immersed in 20% acetic acid for gas foaming and extensively washed in water. Finally, scaffolds were freeze-dried as previously described [47]: scaffolds were frozen at −20 °C, then dried a first time under low pressure (0.010 mbar) at −5 °C and a second time at 30 °C. Scaffolds were then stored at room temperature. One percent fluorescein isothiocyanate (FITC)-dextran was added to the solution as a fluorescent tracer for confocal microscopy. For functionalization studies, we used laminin from Engelbreth-Holm-Swarm murine sarcoma basement membrane (Sigma-Aldrich), commonly used for mouse cell cultures. Some of the freeze-dried Matrix 1% formulations were simply impregnated with laminin (Matrix 1% L) in solution at 100 µg/mL, (Sigma-Aldrich) or with laminin plus another freeze-drying step (Matrix 1% L-FD).

### 4.2. Scaffolds Characterization

#### 4.2.1. Morphology and Porosity

Structure of the dried polysaccharide-based scaffolds was analyzed by scanning electron microscopy (SEM) using a JOEL CarryScope after sputtering with gold. Samples were observed in secondary electron mode at an accelerating voltage of 20 kV. Scaffolds prepared with 1% FITC-dextran were analyzed by confocal microscopy (Carl Zeiss^®^ LSM 780, 10× objective, 2 × 2 tile scan and Z-stack 70 µm image acquisitions) after hydration in PBS. Porosity was computed with the ImageJ^®^ software.

#### 4.2.2. Phosphorus Content

Phosphorus content of polysaccharide-based scaffolds was quantified according to a colorimetric method [48]. About 2 mg of each scaffold was degraded in 1 mL of 10% nitric acid at 105 °C for 3 h. Subsequently, 0.4 mL of 14.7 M nitric acid, 2 mL of 10 mM ammonium metavanadate and 2 mL of 40 μM ammonium pentaphosphoric acid molybdate were added to scaffold lysates. The phosphorus content was finally determined according to a calibration curve based on phosphoric acid.

#### 4.2.3. Swelling Behavior

Hydrogels swelling behavior was assessed in water and in phosphate-buffered saline (PBS). The freeze-dried scaffolds were weighted before and after hydration at several time points (10 min, 30 min, 1 h, 2 h, 3 h, 4 h, 5 h, 6 h and 24 h). Samples soaked in water or PBS were weighted after removal of excess solvent. To determine the swelling ratio, the following equation was used: Sw = (Ws − Wi)/Wi [49], where Sw is the swelling ratio, and Ws and Wi are sample weights in swollen and dry states, respectively.

#### 4.2.4. In Vitro Degradation

Four samples of each formulation were investigated for in vitro enzymatic degradation. After complete hydration in PBS, samples were incubated in a pullulanase and dextranase mixture (10% and 5% *v*/*v* respectively) at 37 °C. The immersed samples were weighted every 5 min until no relevant mass loss was observed. At each time point, the percentage of residual mass (Wt) was calculated according to the following equation: Wt = (Wa/Wb) × 100; Wb is the mass of the scaffolds before degradation, and Wa is the residual mass at each time. Degradation rate was calculated using the slope of the degradation curves.

#### 4.2.5. Rheological Study

Hydrogel disks of 4 cm diameter and about 1 mm height were prepared and hydrated in PBS for mechanical testing. Shear oscillatory measurements were performed on a Discovery HR2 (TA Instrument) equipped with a stainless steel 40 mm diameter crosshatched geometry. Both base and geometry surfaces were rough in order to avoid sample slipping during acquisitions. Axial force was defined at 0.2N for all measurements. First, the linear viscoelasticity domain was determined along a deformation from 0.01% to 10% (data not shown). Then storage (G’) and viscous (G″) moduli were recorded according to a frequency range of 0.05–5 Hz at a fixed deformation of 0.1%. The average value of the storage and loss moduli was measured at least three times and is given here at 1 Hz frequency.

### 4.3. Culture of Mouse Embryonic Neurons

#### 4.3.1. Ethical Approval and Animal Management

All mouse lines were maintained in a mixed C57BL6/DBA2 background. We used the following alleles or transgenes as indicated in the original publications: *PGK^Cre^* [50], *Rosa26^tdTom^* [35]. Day of the plug was considered E0.5. All animal manipulations were approved by a French Ethical Committee (Project Ce5/2016/3996) and were performed according to French and European Union regulations.

#### 4.3.2. Isolation and Culture of Mouse Embryo Cortical Neurons

Neocortex was carefully dissected from E15.5/E17.5 *PGK^Cre^, Rosa26^tdTomato^* embryos and digested in EBSS with 20 U/mL papain, 0.005% DNase (Papain dissociation kit, Worthington) for 25 min at 37 °C. Samples were then mechanically dissociated in EBSS with 0.005% DNase and cells were resuspended at 5 × 10^4^ cells per µL in Neurobasal medium (Life Technologies), supplemented with 1 × SM1 supplement (Stem Cell), 200 mM L-Glutamine and 1× penicillin–streptomycin (both from Life Technologies). Cells were seeded on scaffolds as previously described [27,51]: (i) scaffolds were placed in a syringe along with a cell suspension of approximately 3 × 10^5^ cells; (ii) the plunger was introduced, and the syringe tip was closed using a 3-way valve; finally (iii) vacuum was induced by moving the plunger about 3 cm up and down until the scaffold was fully impregnated and became transparent. Scaffolds were then placed in 12-well plates. Culture medium was replaced after five days by BrainPhys supplemented with 1X SM1 supplement (both from Stem Cell).

#### 4.3.3. Immunohistochemistry and Imaging

Scaffolds were fixed in 4% paraformaldehyde (PFA) in phosphate-buffered saline solution (PBS) at room temperature (RT) for 15 min and then washed and stored in PBS at 4 °C. The following immunostaining protocol was used. Scaffolds were incubated for 1 h at room temperature with 10% donkey serum, 0.25% Triton X-100 in PBS. Primary antibodies were incubated in the same solution overnight at 4 °C, and secondary antibodies were incubated for 2 h at room temperature in 1% donkey serum, 0.25% Triton X-100 in PBS solution. Scaffolds were counterstained with Hoechst (H3570, Life technologies) for nuclei detection. Primary antibodies were used at the following dilutions: rabbit anti-Tuj1 (BioLegend, 1:1000) and goat anti-mCherry (Sicgen, 1:500). Secondary antibodies were from Jackson Immuno Research. Scaffolds were mounted in 1× PBS under a coverslip for analysis, and optical sections were obtained on a confocal microscope (SP5, Leica). The ImageJ software was used to generate Z-stacks and assemble pictures.

#### 4.3.4. Cell Quantification

Four to height scaffolds (4 mm diameter) of each formulation were used to count cells after one, four and seven days of culture. Scaffolds were digested in a pullulanase and dextranase mixture (10% and 5% *v*/*v* respectively) at 37 °C for 25 min and then mechanically dissociated. Cell count was performed using a Countess II automated cell counter (Invitrogen).

### 4.4. Statistical Analysis

All results are presented as mean ± standard deviation (SD). All experiments were performed at least in triplicate. GraphPad Prism^®^ 5.0 software was used to perform statistical analysis using one-way ANOVA test with Tukey post-test for phosphorus content and degradation rate, two-way ANOVA test with Bonferroni post-test for swelling ratio, degradation kinetics and Mann–Whitney post-test for rheology measurement. Statistical significance is denoted as * *p* < 0.05; ** *p* < 0.01; *** *p* < 0.001.

## 5. Conclusions

The goal of this study was to determine the potential of tailored macroporous polysaccharide-based scaffolds for the 3D culture and manipulation of cortical neurons isolated from embryonic mouse cortex. For this, we prepared and tested several formulations with different STMP concentrations (1%, 1.5% and 3%), with or without laminin functionalization. After assessing their mechanical properties and in vitro biodegradability, we explored their ability to promote the attachment and maturation of embryonic neurons. We showed that softer scaffolds with 1% STMP (*w*/*w*) functionalized with freeze-dried laminin better supported neuronal attachment and neurite outgrowth. These scaffolds are a promising lead for further research in the field of 3D neural cell culture, with potential for use in in vitro and in vivo studies.

## Figures and Tables

**Figure 1 ijms-22-12726-f001:**
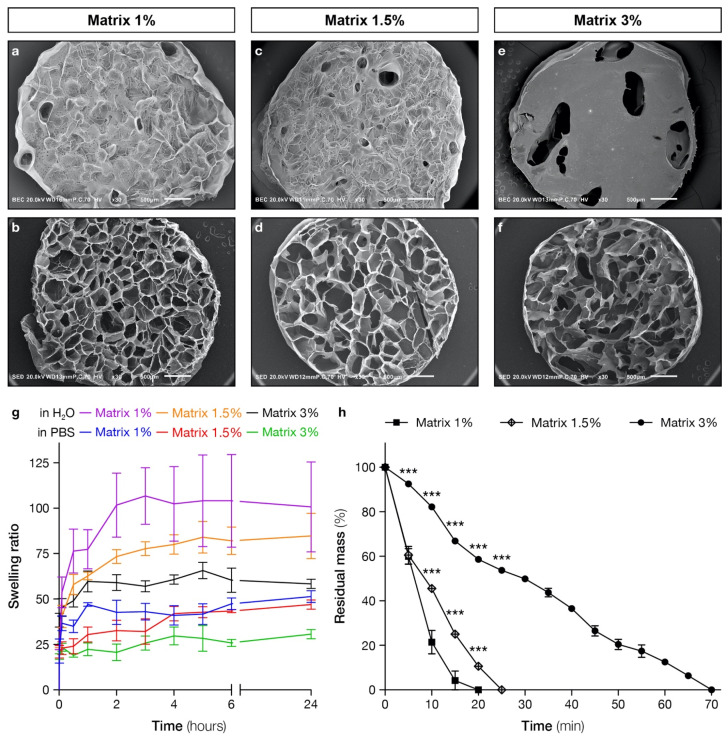
Scaffolds morphology and structure. (**a**–**f**) Scanning electron micrographs of pullulan/dextran scaffolds crosslinked with 1%, 1.5% and 3% (*w*/*w*) of STMP. Top view of either scaffolds surface (**a**,**c**,**e**) or section (**b**,**d**,**f**). Scale bars: 500 μm. (**g**) Swelling kinetic of pullulan/dextran scaffolds prepared with 1%, 1.5% and 3% (*w*/*w*) of STMP. Samples were hydrated in PBS or water until maximum hydration was reached. Results are presented as mean values ± SD from dried state. Three samples were analyzed for each formulation. (**h**) In vitro enzymatic degradation kinetics of scaffolds prepared with 1%, 1.5% and 3% (*w*/*w*) of STMP, the cross-linking agent. Once incubated in the pullulanase/dextranase solution, samples were weighted every 5 min to measure the residual mass. Four samples were analyzed for each time point. *** *p* < 0.001.

**Figure 2 ijms-22-12726-f002:**
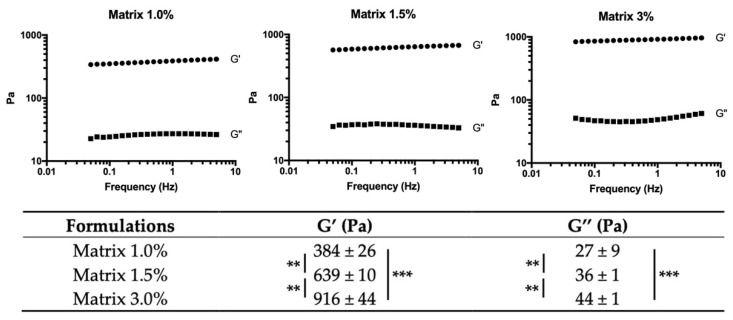
Rheological characterization. Elastic (G′) and viscosity (G″) moduli of hydrated pullulan/dextran scaffolds crosslinked with 1%, 1.5% and 3% (*w*/*w*) of STMP. Three samples were analyzed for each formulation. *** *p* < 0.001, ** *p* < 0.005.

**Figure 3 ijms-22-12726-f003:**
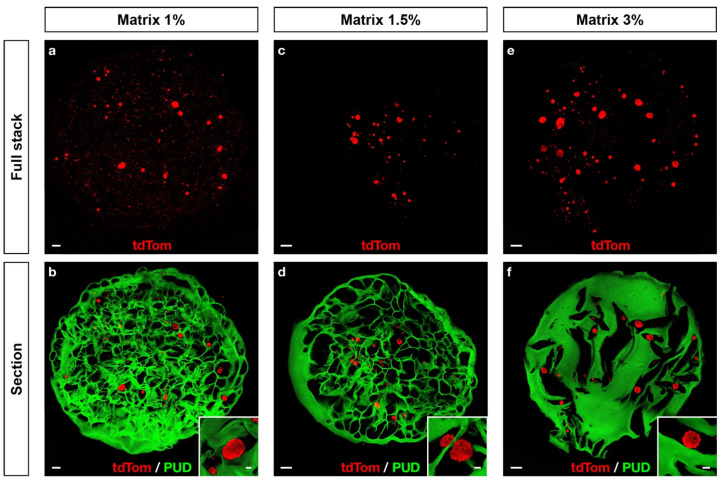
Embryonic neurons seeded in polysaccharide-based scaffolds form aggregates. Mouse embryonic neurons (red) were seeded and cultured for two days on FITC-traced scaffolds (green) prepared with 1% (**a**,**b**), 1.5% (**c**,**d**) and 3% (**e**,**f**) (*w*/*w*) of STMP. Maximum Z-stack projections (**a**,**c**,**e**) give an estimation of cell density and morphology. Simple Z-sections (**b**,**d**,**f**) include FITC scaffold tracing for morphology. Scale bars: (**a**–**f**), 200 μm; inserts, 30 μm.

**Figure 4 ijms-22-12726-f004:**
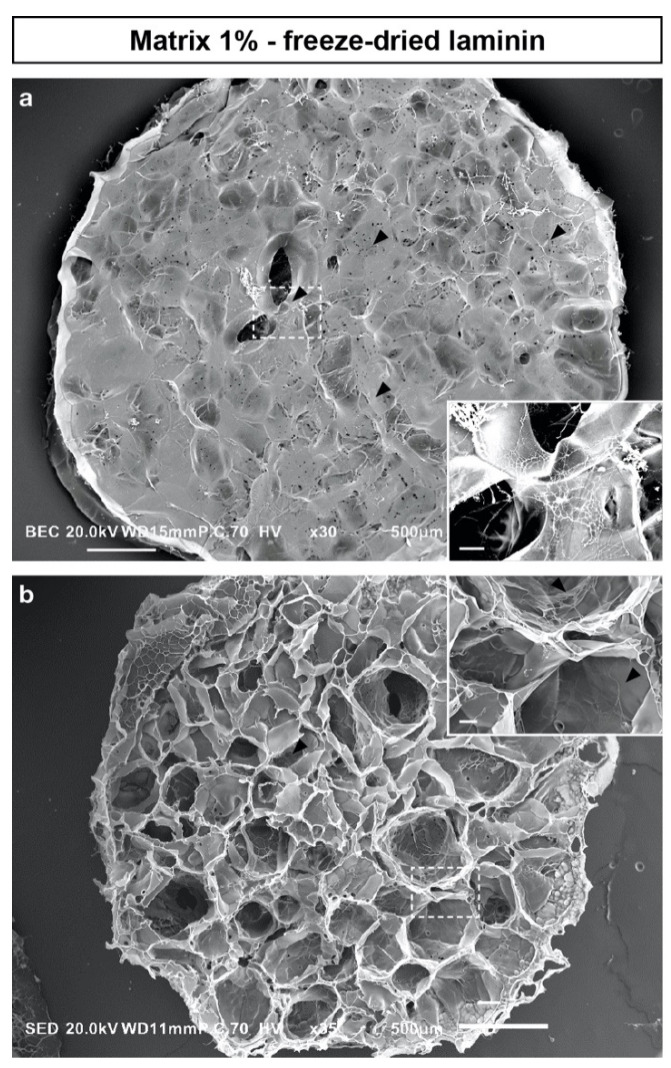
Freeze-dried laminin creates a mesh network within the scaffold. Scanning electron micrographs of a pullulan/dextran scaffold prepared with 1% (*w*/*w*) of STMP, hydrated with a laminin solution (100 μg/mL) and freeze-dried. Top view either of the scaffold surface (**a**) or section (**b**). Arrowheads point to the laminin filaments (in white). Scale bars: (**a**,**b**), 500 μm; insert, 50 μm.

**Figure 5 ijms-22-12726-f005:**
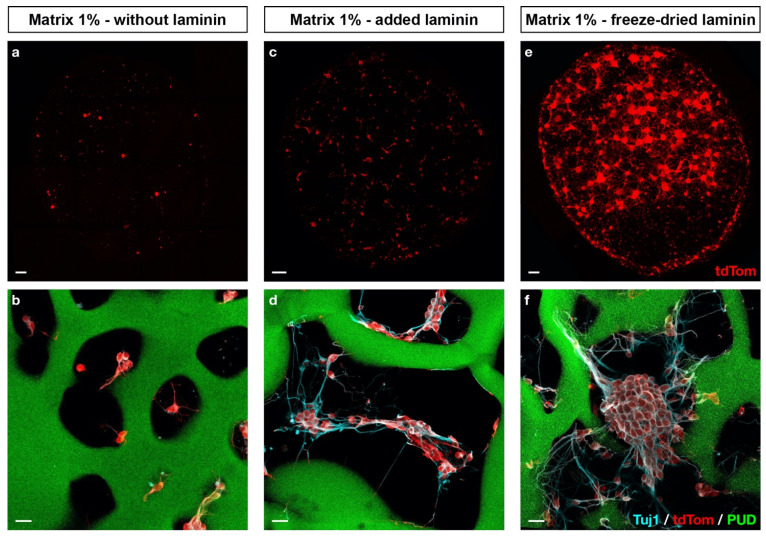
Freeze-dried laminin improves cell adhesion to the scaffold. Mouse embryonic neurons (red) were seeded and cultured for two days on FITC-traced scaffolds (green) prepared with 1% (*w*/*w*) of STMP, either without laminin (**a**,**b**), with added laminin (100 μg/mL) during cell seeding (**c**,**d**) or with freeze-dried laminin before cell seeding (**e**,**f**). Maximum Z-stack projections (**a**,**c**,**e**) give an estimation of cell density and morphology. Higher magnification pictures (**b**,**d**,**f**) include immature neuron marker βIII-tubulin (Tuj1, cyan) and FITC scaffold tracing (green). Scale bars: (**a**–**f**), 200 μm; inserts, 30 μm.

**Table 1 ijms-22-12726-t001:** Physicochemical characterization of 1%, 1.5% and 3% matrices. Phosphorus content of pullulan/dextran scaffolds crosslinked with 1%, 1.5% and 3% (*w*/*w*) of STMP is presented as mean values ± standard deviation (SD) from dried state. Four samples were analyzed for each formulation. Degradation rates were calculated using the slope from the in vitro degradation curves of scaffolds prepared with 1%, 1.5% and 3% (*w*/*w*) of STMP. Once incubated in the pullulanase/dextranase solution, samples were weighted every 5 min to measure the residual mass. Four samples were analyzed for each experiment. *** *p* < 0.001.

Formulations	Phosphorus Content (μmol/g)	Degradation Rates (mg/min)
Matrix 1.0%	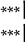	84 ± 1	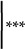	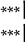	5.11 ± 0.14	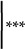
Matrix 1.5%	108 ± 5	3.83 ± 0.12
Matrix 3.0%	236 ± 7	1.39 ± 0.03

## Data Availability

The data presented in this study are available on request from the corresponding author.

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
