# Peer review of "Tuning Physicochemical Properties of a Macroporous Polysaccharide-Based Scaffold for 3D Neuronal Culture"

_ijms, 2021, doi:10.3390/ijms222312726_

Round 1

Reviewer 1 Report

Dear authors,

Here are my comments for this paper:

  1. You must clearly show the novelty of your paper with respect to existent literature.
  2. Please show swelling curves for your materials. What about the swelling kinectics which is very important?
  3. Please show plots of G' and G'' vs frequency.
  4. You should use units for MW (g/mole or Da).
  5. Why did you choose both water and PBS for swelling studies?
  6. Add conditions for freeze-drying process.
  7. Why did you use pullulan and dextran in this combination? You mentioned some advantages, but the idea must be strenghten.

Reviewer 2 Report

Manuscript ijms-1451580 describes the preparation of a pullulan/dextran polysaccharide porous scaffold and its utilization for 3D neuronal culture.

Lamin is well-known to enhance cell adhesion on different substrates. Laminin-modified gellan gum hydrogels were proven to improve the proliferation and differentiation of neuronal stem cells –Li et al. 2020, Laminin-modified gellan gum hydrogels loaded with the nerve growth factor to enhance the proliferation and differentiation of neuronal stem cells. RSC Advances, 10(29), 17114-17122. In the Discussion Section, I suggest the authors also mention this paper.

 L70-L78. This paragraph is more appropriate to the Conclusion Section.  I suggest the authors to replace it with a short presentation of the goal and objective of the paper.

Reviewer 3 Report

Manuscript No. ijms-1451580

„Tuning physicochemical properties of a macroporous polysaccharide-based scaffold for 3D neuronal culture” by Gaspard Gerschenfeld, Rachida Aid, Teresa Simon-Yarza, Soraya Lanouar, Patrick Charnay, Didier Letourneur, Piotr Topilko for International Journal of Molecular Sciences

Comments:

  1. Were the cells just added to the material being tested or was there any mechanical attempt to insert the cells into the pores of the material, e.g. by gentle stirring or shaking? How did the authors ensure that toxic effect on cells did not occurred in the pores of the carrier. How could we ensure that the cells in the pores of the carrier without mechanical agitation were not poisoned?
  2. What was the viability of the cells added to the material and after cell recovery?
  3. No figures were provided for the number of cells before and after culturing on the carrier. It is very important from the point of view of obtaining the appropriate number of cells, e.g. in a clinical setting.
  4. Was the doubling time of the cell population analyzed by selecting the incubation time? When studying other types of cells, will they only persist in the pores of the carrier or will be stimulated to proliferation?
  5. Was its cross-linking taken into account when preparing the biomaterial, adjusting it to the size of the tested cells?
  6. Have Authors also tried to test human cells (cell lines)? Can the authors in the discussion refer to the possibility of culturing human cells not only neuronal but also glial? The analysis of the neurons themselves is not enough to be able to draw far-reaching conclusions.
  7. Without answers to the above questions in the manuscript, the conclusion of the work is not fully justified. It should then be presented as merely proposing a modified biomaterial for research and should be corrected.

Round 2

Reviewer 1 Report

Dear authors,

The paper is now improved after the revision. Nevertheless, the swelling kinetics is not shown. Maybe you did not evaluate the kinetics (diffusion control, chain relaxation control or others). You should check this aspect.

Reviewer 3 Report

The manuscript is sufficiently improved.